# Recurrent Bayesian Classifier Chains for Exact Multi-Label Classification

**Walter Gerych, Thomas Hartvigsen, Luke Buquicchio, Emmanuel Agu, Elke Rundensteiner**
Worcester Polytechnic Institute
Worcester, MA
{wgerych, twhartvigsen, ljbuquicchio, emmanuel, rundenst}@wpi.edu

## Abstract

Exact multi-label classification is the task of assigning each datapoint a set of class labels such that the assigned set exactly matches the ground truth. Optimizing for exact multi-label classification is important in domains where missing a single label can be especially costly, such as in object detection for autonomous vehicles or symptom classification for disease diagnosis. Recurrent Classifier Chains (RCCs), a recurrent neural network extension of ensemble-based classifier chains, are the state-of-the-art exact multi-label classification method for maximizing subset accuracy. However, RCCs iteratively predict classes with an *unprincipled* ordering, and therefore indiscriminately condition class probabilities. These disadvantages make RCCs prone to predicting inaccurate label sets. In this work we propose Recurrent Bayesian Classifier Chains (RBCCs), which learn a Bayesian network of class dependencies and leverage this network in order to condition the prediction of child nodes only on their parents. By conditioning predictions in this way, we perform principled and non-noisy class prediction. We demonstrate the effectiveness of our RBCC method on a variety of real-world multi-label datasets, where we routinely outperform the state of the art methods for exact multi-label classification.

## 1 Introduction

Exact multi-label classification (E-MLC) is the task of assigning multiple labels to a single instance, and is critical for a variety of of domains. For example, when detecting health conditions, one person will often experience *multiple* symptoms concurrently. Additionally, missing the detection of even one symptom can have drastic effects and so the entire label set must be predicted *exactly* correct. Missing the symptom *Loss of Taste*, for instance, might cause a downstream classifier to misdiagnose COVID-19 as the common cold [3]. While explicitly optimizing for exact classification is inordinately challenging, a successfully-trained model is highly rewarding as it is guaranteed to capture the joint probabilities between the features and all labels.

There has been increasing interest in developing multi-label classification methods for E-MLC [17, 19, 18, 13, 7, 28, 15, 4, 24]. Most are based on *classifier chains*, which are ensembles of models where each classifier predicts a single class and the predictions from the $i$-th model are used as input into the $i + 1$-th model in order to learn inter-class dependencies. As the classifier chain requires an ordering of the classes *a priori* [19], there are variations of classifier chains that utilize different class orderings [11, 16], such as *Bayesian classifier chains* (BCCs) [30, 23] which utilize a Bayesian network to learn class orderings. A key disadvantage of these classier chain approaches is that they all require an ensemble of individual classifiers both requiring a large number of parameters and learning a different and independent latent representation of the input data per classifier. This makes such ensemble methods particularly prone to overfitting [20, 1].

35th Conference on Neural Information Processing Systems (NeurIPS 2021).

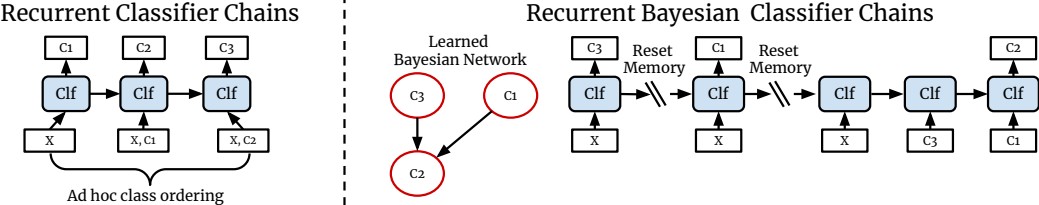

Figure 1: A comparison of the difference between Recurrent Classifier Chains and our proposed *Recurrent Bayesian Classifier Chain*s.

The leading extension of classifier chains is the *recurrent classifier chain* (RCC), which models inter-class dependencies by using a recurrent network to iteratively predict each class, such that each class prediction is conditioned on the predictions of all previously predicted classes. The order in which classes are predicted is likewise important for RCCs [19, 15]. To address this, the ordering is usually ad hoc with popular choices being to predict classes either frequent-to-rare or rare-to-frequent [15], while some very recent RCCs attempt to learn an ordering for the classes [4, 24]. However, there is no direct feedback on the optimal class ordering, and attempting to learn through reinforcement alone is a challenging task that is also prone to overfitting [29]. Additionally, RCCs inherently condition their class predictions on *noisy inputs*, since the predictions are conditioned on all previously-predicted classes *even when most classes are independent of all other classes*.

The task of E-MLC is to train a model $f$, such that $f$ can predict the *exact* label sets of previously-unseen instances. Specifically, our goal is to use training data to obtain a classifier that can maximize the *Subset Accuracy* of unseen testing data, as the label set most likely to be *exactly* correct is the risk minimizer for Subset Accuracy. Thus, maximizing Subset Accuracy directly corresponds to optimizing for E-MLC. Subset Accuracy is a strict multi-label metric that returns $1$ if the entire label set is correct, and $0$ otherwise.

There are four major challenges for E-MLC:

- *Critical label dependencies*. Inter-class dependencies must be learned and modeled in order to perform E-MLC [7].
- *Cascading class errors*. When class predictions are made sequentially, early mistakes are likely to lead to additional errors for subsequent classes.
- *Strict metric*. Subset Accuracy is only nonzero if the predicted label set is *exactly* correct, which means that a mostly-correct prediction is penalized as harshly as a completely incorrect solution during evaluation. This non-smooth metric is naturally challenging to maximize.
- *Exponential label set growth*. The number of possible label sets grows exponentially with the number of classes. Thus, finding the label set that is *exactly* correct is akin to finding a needle in a haystack for large numbers of classes.

We propose the *Recurrent Bayesian Classifier Chain* (RBCC), the first recurrent classifier chain that avoids conditioning on noisy inter-class relationships, thereby learning a highly-*principled* ordering for class predictions. The RBCC consists of three core components. First, the RBCC explicitly learns inter-class dependencies using the *Inter-class Dependency Network*, which finds a Bayesian network that models the relationships between the classes. Second, this network structure is used by the *Recurrent Conditional Dependency Model*, which for each class produces a latent vector representation that summarizes the input features along with the classes upon which it depends. This representation is then passed on to the *Bayesian-Conditioning Classifier*, which produces the probability of observing each class. Importantly, the Bayesian-Conditioning Classifier's predictions for a given class are *only* conditioned on the input features and the classes for which it is not conditionally independent (which we refer to as the *parent* classes). Altogether, the RBCC thus produces accurate label sets, optimizing for exact multi-label classification by mitigating cascading errors.

**Contributions.** The contributions of this work are as follows:

- The RBCC is the first recurrent classifier chain method that utilizes conditional independence to perform non-noisy class conditioning.

- We propose a novel training and inference algorithm for this approach.
- Experimentally, we show that RBCCs significantly outperform state-of-the art methods for E-MLC on a variety of real-world datasets.

## 2 Methodology

### 2.1 Problem Definition

Formally, we define the problem of E-MLC as follows: Let $(X, C_1, ..., C_L) \sim P_{X \times C_1 \times ... \times C_L}$, where $X$ is the random variable of features and $C_i$ is the $i$th class, such that there are $L$ possible classes. If the observed value of $C_i$, $c_i$, is 1, then the $i$th class applies. Otherwise, $c_i = 0$ and the class does not apply. Let $\mathbb{D} = \{(\boldsymbol{x}, \boldsymbol{y})_i\}_{i=1}^{N}$ be a dataset of $i.i.d$ samples from $P_{X \times C_1 \times ... \times C_L}$, such that each $\boldsymbol{x} \in \mathbb{R}^m$ is an observed value of $X$ and $\boldsymbol{y}$ is a vector of length $L$ of observed values for all classes, and thus $[\boldsymbol{y}]_i$ (the $i$th element of vector $\boldsymbol{y}$) is equal to $c_i$. Our goal is to find a classifier $f_\theta$ that maximizes the *Subset Accuracy* of unobserved test data. The Subset Accuracy ($SA$) of a prediction $f_\theta(\boldsymbol{x})$ is defined as:

$$SA(f_\theta(\boldsymbol{x}), \boldsymbol{y}) := [\![\boldsymbol{y} = f_\theta(\boldsymbol{x})]\!],$$

where $[\![\cdot]\!]$ is the conditional function such that $[\![P]\!]$ returns 1 if the predicate $P$ is true, and is 0 otherwise.

The risk minimizer for SA is given by:

$$h^*(\boldsymbol{x}) = \arg\max_{\boldsymbol{y}} \boldsymbol{P}(C_1 = [\boldsymbol{y}]_1, ..., C_L = [\boldsymbol{y}]_L | X = \boldsymbol{x}) \tag{1}$$

Thus, our classifier $f_\theta$ should aim to return the Maximum A Posteriori of the class probabilities given observations $\boldsymbol{x}$.

### 2.2 Proposed method: RBCC

We propose the *Recurrent Bayesian Classifier Chain* (RBCC), which improves upon RCCs by performing class prediction in a *principled order* with *non-noisy* class conditioning during prediction. We achieve this in part by learning a Bayesian network representation of inter-class dependencies. We note here that the focus of this work is *not* on a new method to construct a Bayesian network of class inter-dependencies, of which there are many existing methods [22, 6, 22, 27, 5]. Instead, the focus is on how to better incorporate a Bayesian network found through these existing methods into the recurrent classifier chain system.

At a high level, RBCC aims to find to find the parameters of a classification model such that the predicted label vector is the most likely to *exactly* match the true label vector $\boldsymbol{y}$. Thus, we are aiming to return the risk minimizer given in Equation 2, and are as such optimizing for the exact goal of E-MLC.

This is achieved through finding a model of $\boldsymbol{P}(C_1, ..., C_L | X)$ parameterized by $\theta$:

$$\theta^* = \arg\max_{\theta} \sum_{i=1}^{N} log \boldsymbol{P}_\theta(\boldsymbol{x}_i; \theta). \tag{2}$$

Then, the prediction for $\boldsymbol{y}_i$ can be found be searching for the maximum of $\boldsymbol{P}_\theta(\boldsymbol{x}_i; \theta)$.

At a glance, one to model $\boldsymbol{P}(C_1, ..., C_L | X)$ would be to factorize it into a product of per-class probabilities:

$$\boldsymbol{P}(C_1, ..., C_L | X) = \boldsymbol{P}(C_1 | \boldsymbol{x}) \prod_{i=2}^{L} \boldsymbol{P}(C_i | C_{<i}, \boldsymbol{x}), \tag{3}$$

as is done by [15]. However, there are two issues apparent with the above factorization. First, while in theory we can order the classes in the above product in any way *if the exact true conditional probabilities are known*, in practice the order of classes has an immense effect on classification performance [19]. For instance, if $C_m$ is strongly dependent class $C_n$, then the prediction for $c_m$ should be conditioned on that class during training for more accurate predictions. However, depending on the ordering of the classes, $c_m$ may not be conditioned on $c_n$ in the factorization above. Second,

| Symbol | Meaning |
|--------|---------|
| $X$ | Feature random variable |
| $C_k$ | Random variable of $k$th class |
| $c_k$ | Observed value for $k$th class. If $c_k = 1$ the class applies; otherwise, $c_k = 0$ and the class does not apply |
| $\boldsymbol{x}$ | Observed feature value for particular instance |
| $\boldsymbol{y}$ | Vector of class observations; $\boldsymbol{y} = [c_1, c_2, ..., c_L]$ |
| $[(\cdot)]_k$ | $k$th entry in vector $(\cdot)$ |
| $\mathcal{G}$ | Bayesian network of class dependencies |
| Unbolded capital letter | A random variable |
| Unbolded lowercase | Observed value of random variable |
| **Bold** uppercase letter | A matrix |
| **Bold** lowercase letter | A vector |

Table 1: Meaning of the most import notation used in this work.

each class $C_k$ is conditioned on *all* of the preceding classes $C_{<k}$, even if $C_k$ is independent from most of these classes. Approaches that use the factorization shown in Equation 3 condition the prediction of $c_k$ on noise, risking overfitting to false correlations between classes during training.

We can solve these two issues by assuming that the joint dependencies among all classes $C_1, C_2, \ldots, C_\ell$ and features $X$ can be factorized as a directed acyclic graph (DAG) $\mathcal{G}$. Each node in the graph corresponds to either a class $C_i$ or the features $X$, and the edges encode the dependencies between nodes. As we make the standard multi-label assumption that all classes depend on the input features $X$, $X$ is a parent of all other nodes in $\mathcal{G}$, other than itself. Let $Pa_{\mathcal{G}\backslash X}(C_k)$ be the parents of $C_k$, other than the universal parent $X$. We can thus simplify $\boldsymbol{P}(C_1, ..., C_L|X)$ into the following:

$$\boldsymbol{P}(C_1, C_2, \ldots, C_L|X) = \prod_{i=1}^{L} \boldsymbol{P}(C_i|Pa_{\mathcal{G}\backslash X}(C_k), X). \tag{4}$$

The above equation does not have the flaws of the factorization in Equation 3, as each class is now *only* conditioned on *all* of the classes upon which it depends.

RBCC thus involves three primary steps to encode this key idea into one unified multi-label model: For each class we need to first identify its parent classes, summarize the information of the observations of the parent classes, and then make a prediction for that class based on the observations of its parents.

In the first step, we learn the structure of the inter-class dependencies using the *Inter-class Dependency Network*, which fits a Bayesian network to the classes. This network does not tell us the strength of the dependencies between classes and does not alone allow us to infer which classes are associated with a given instance. Instead, the learned network identifies which classes a given class is dependent on (the *parent classes*), and which classes it is independent of given the parent classes. This allows us to condition a down-stream classification model on *only* the required parent classes of each class, reducing cascading errors from incorrect class predictions.

The input features and observations of the parent classes are summarized by the *Recurrent Conditional Dependency Model*, a recurrent neural network-based architecture, before this summarized representation is passed into the down stream classifier, which we refer to as the Bayesian-Conditioning Classifier. The Bayesian-Conditioning Classifier model makes predictions for each class $C_k$ such that the predictions are conditioned only on the observed features $\boldsymbol{x}$ and the parent classes of $C_k$, according to the network structure learned from the Inter-class Dependency Network.

### 2.2.1 Inter-class Dependency Network.

The Inter-class Dependency Network learns for each class both which other classes it depends on and which classes it is independent from *when conditioned on* $X$. Thus, for class $C_k$, the Inter-class Dependency Network finds $Pa(C_k)$, the set of classes that such that $\boldsymbol{P}(C_k|X) \neq \boldsymbol{P}(C_k|C_\ell, X) \; \forall \; C_\ell \in Pa(C_k)$.

We accomplish this by assuming that the inter-class dependencies can be modeled by a *Bayesian network*. Thus, the inter-class dependencies are modeled by a directed graph structure such that each node represents a class, and the edge from the node for $C_\ell$ to $C_k$ means that $C_\ell \in Pa(C_k)$. Furthermore, the Bayesian network is by definition *acyclic*, implying that if $C_\ell \in Pa(C_k)$, then $C_k \notin Pa(C_\ell)$. This implies that we can obtain a principled ordering for class predictions, such that each class can always be predicted after its parent classes.

Note that we are interested in the parents of each class *conditioned on the features* $X$. This is an important distinction, because the structure of the inter-class dependencies may differ from the structure of the classes when conditioned on the observed features. Thus we can not learn the desired conditional class inter-dependencies by simply fitting a Bayesian network structure to the classes.

To model the correct conditional dependencies, we follow the approach of *Zhang et al.* [30] and identify the *residuals* of each class. To this end, we assume that the relationship between each class and $X$ can be expressed as

$$C_k = g_k(X) + E_k,$$

such that $g_k(X)$ is the component of $C_k$ that is predictable from $X$, and $E_k$ is the *irreducible error* of $C_k$ that is not predictable from $X$. Note that if there are inter-class dependencies, $E_k$ *will* be in part predictable given some other class.

**Lemma 1.** *The Bayesian network structure of the classes conditioned on $X$ is the same as the network structure of $\{E_1, E_2, \ldots, E_L\}$.*

Lemma 1, proven in [30], implies we can learn the inter-class dependency structure by fitting a Bayesian network to the residuals of each class. For each class $C_k$ we train a classifier $g_k : \mathbb{X} \to \{0, 1\}$ which predicts $c_k$ given $\boldsymbol{x}$. We then obtain $E_k$ as the residual of the predictions of $g_k$.

The final step is to fit a Bayesian network to the residuals. As finding the Bayesian network structure is NP-hard, we must estimate this structure. There are many algorithms to approximate fit Bayesian networks [22, 6, 27, 5]. We use *hill climbing* [22] for our experiments due to its low computational cost. After fitting the Bayesian network, we obtain the graph structure $\mathcal{G}_E$ and determine the parents $Pa_{\mathcal{G}_E}(E_k)$ for every $E_k$. Note that from this we can obtain $Pa_{\mathcal{G}\backslash X}(C_k)$ for each $C_k$, because if $E_i \in Pa_{\mathcal{G}_E}(E_k)$ then $C_i \in Pa_{\mathcal{G}\backslash X}(C_k)$.

### 2.2.2 Recurrent Conditional Dependency Model.

With knowledge of each class's parents in hand, we next perform classification (i.e., we want to find $\boldsymbol{P}(C_k = 1|Pa_{\mathcal{G}\backslash X}, X = \boldsymbol{x})$ for each class $C_k$). However, the number of parents will vary for each class. This implies that class prediction must be performed with a variable number of inputs. This remains a standing challenge for state-of-the-art machine learning, as most methods require constant input sizes. For this reason, we leverage recent successes in representation learning, summarizing the input features $\boldsymbol{x}$ and the observed valuyes of the set of parent classes $Pa_{\mathcal{G}\backslash X}(C_k)$. This is achieved using our proposed Recurrent Conditional Dependency Model.

The Recurrent Conditional Dependency Model utilizes a recurrent network that learns a joint feature representation that summarizing the input features and observed parent classes for each class $C_k$ of a given instance. This representation is learned by reading in the observed features and parent classes sequentially. To mitigate effects of vanishing gradients, we compute use a *gated recurrent unit* (GRU) for the recurrent network to create the representation. Thus, the representation $a_k$ is determined by:

$$\boldsymbol{r}_t = \sigma(\boldsymbol{W_r v} + \boldsymbol{U_r h}_{t-1} + \boldsymbol{b}_r)$$
$$\boldsymbol{z}_t = \sigma(\boldsymbol{W_z v} + \boldsymbol{U_z h}_{t-1} + \boldsymbol{b}_z)$$
$$\boldsymbol{s}_t = \Phi(\boldsymbol{W_a v} + \boldsymbol{U_a}(\boldsymbol{r}_t \odot \boldsymbol{a}_{k,t-1}) + \boldsymbol{b}_a)$$
$$\boldsymbol{a}_{k,t} = \boldsymbol{z}_t \odot \boldsymbol{a}_{k,t-1} + (1 - \boldsymbol{z}_t) \odot \boldsymbol{s}_t,$$

such that $\boldsymbol{W}_{r,z,a}$ and $\boldsymbol{U}_{r,z,a}$ are the weight matrices of the GRU, and $\boldsymbol{v} = <\boldsymbol{x}, enc(c_t)>$, where $<a, b>$ is the concatenation of $a$ and $b$ and $enc(c_t$ is a vector encoding of $c_t$. $\sigma$ is the sigmoid function and $\phi$ is the hyperbolic tangent function. In general, the choice of recurrent architecture is modular and can be substituted with some other architecture, such as an LSTM.

The above process is repeated for all $C_t$ in $Pa_{\mathcal{G}\setminus X}(C_k)$. The final representation $\boldsymbol{a}_k$ for class $C_k$ is obtained after reading in the observed value for the final parent class. If the class $C_k$ has no parents, then we set $\boldsymbol{v} = <\boldsymbol{x}, \boldsymbol{0}>$ and compute the above equations only once each.

### 2.2.3  Bayesian-Conditioning Classifier.

We can now use the summarized information of the parent classes and input features to condition a classifier to produce $\boldsymbol{P}(C_k|Pa(C_k), X)$ for each class $C_k$, as is required for Equation **??**. We model this probability using the Bayesian-Conditioning Classifier.

We thus pass said representation $\boldsymbol{a}_k$ into a feed forward network $m$, such that

$$m(\boldsymbol{a}_k) = \sigma(\boldsymbol{W}_H \boldsymbol{a}_k) + \boldsymbol{b}_H,$$

where $\boldsymbol{W}_H \in \mathbb{R}^{L,|\boldsymbol{a}_k|}$ and $\boldsymbol{b}_H \in \mathbb{R}^L$ are the weight and bias vector of $m$, respectively. Thus, $m$ maps $\boldsymbol{a}_k$ into a vector of length $L$, where $[m(\boldsymbol{a}_k)]_k$ is our model of $\boldsymbol{P}(C_k = 1|Pa(C_k), X = \boldsymbol{x})$.

To avoid inadvertently conditioning on noise by conditioning our prediction for a given class $C_k$ on classes other than its parents, we reset the hidden representation $\boldsymbol{a}_k$ after predicting each class. Thus, we perform *many-to-one* prediction for each class, where we read in the sequence of parents and then predict a binary value for the class. This differs from the approach taken by standard RCCs, as they perform *many-to-many* classification by reading in a sequence of previously predicted class and predicting a sequence of successive classes.

This strategy poses a challenge when multiple classes have the exact same parents. For instance, in if $Pa_{\mathcal{G}\setminus X}(C_k) = Pa(C_\ell)$, then $\boldsymbol{a}_k = \boldsymbol{a}_\ell$. However, we do not necessarily want $\boldsymbol{P}(C_k = 1|Pa(C_k), X = \boldsymbol{x}) = \boldsymbol{P}(C_\ell|Pa(C_\ell), X = \boldsymbol{x})$. For this reason, the feed forward network $m$ that transforms the latent representation into a prediction probability does not map each $\boldsymbol{a}_k$ into $[0, 1]$. Instead, $m$ maps $\boldsymbol{a}_k$ to $[0, 1]^L$, and we take one value of the $L$-dimensional vector (i.e., $[m(a_k)]_k$) to represent $\boldsymbol{P}(C_k = 1|Pa(C_k), X = \boldsymbol{x})$. This way, even if $Ca_k = Ca_\ell$, we can still have $\boldsymbol{P}(C_k = 1|Pa(C_k), X = \boldsymbol{x}) \neq \boldsymbol{P}(C_\ell = 1|Pa(C_\ell), X = \boldsymbol{x})$ as $[m(\boldsymbol{a}_k)]_k$ is not required to equal $[m(\boldsymbol{a}_\ell)]_\ell$.

As all classes that apply to a given instance are observed during training, we can train the Recurrent Conditional Dependency Model and Bayesian-Conditioning Classifier by simply minimizing the binary cross entropy (BCE) between the predicted values for each class and whether or not the class applies to the given instance. Let $\tilde{\boldsymbol{y}} \in [0, 1]^L$ be our probabilistic predicted label set for $\boldsymbol{x}$, such that $[\tilde{\boldsymbol{y}}]_k = [m(\boldsymbol{a}_k)]_k$. Thus, our loss function $\mathcal{L}([\tilde{\boldsymbol{y}}]_k)$ is:

$$\frac{1}{L}\sum_{k=1}^{L} -[\boldsymbol{y}]_k \log([m(\boldsymbol{a}_k)]_k) - (1 - [\boldsymbol{y}]_k)\log(1 - [m(\boldsymbol{a}_k)]_k). \tag{5}$$

During testing, no classes are observed. We thus require an inference strategy that predicts the value of all parent classes before their child classes. To this end, we construct a recursive algorithm that predicts each class given observations for its parent classes. If the parents have not yet been predicted, we recursively call the inference algorithm on the *unpredicted* parent classes.

We note that this results in a greedy estimate of the most likely label vector for each instance. Alternate searches for the most likely label vector, such as an $A^*$ search or $\epsilon$-approximate search, could likewise be performed [19]. We apply greedy search as it is computationally inexpensive and the simplest solution, while still leading to high accuracy as we demonstrate in the following section.

## 3  Experiments

### 3.1  Datasets

We evaluate our method on three commonly used benchmark multi-label datasets [9, 2, 21]; for all of these we use the provided train/test partitions and are not modified.

`PASCAL VOC 2007`[1][9]: This standard multi-label image dataset consists of 9,963 natural images with 20 possible classes. Images were originally collected from "flickr". Each image was featurized by extracting the features of a pretrained ResNet-18 [10] model into 512-dimensional vectors. Specifically, ResNet-18 model was pretrained on ImageNet and the feature representations were extracted from the final average pooling layer.

`Scene`[2][2]: This dataset contains 2407 scenery images with 6 possible labels. The features are 294-dimensional vectors corresponding to the spatial color moments in the LUV space.

`Yelp`[3][21]: This dataset consists of over 10,000 reviews of business submitted by users of Yelp. There are 7 classes, though 3 are mutually exclusive. Each review is featurized into 671 tokens.

The `Scene` and `Yelp` datasets contain no personally identifiable information. `PASCAL VOC` was scraped from Flikr, and therefor might have identifiable information. However, all of the `PASCAL VOC` data was factorized with a non-invertable transformation (Resnet-18 model), and thus no longer will contain identifiable information.

To the best of our knowledge, no explicit consent obtained from the people whose data was collected beyond the terms of service of the websites from which the data was scraped. However, these are standard benchmark datasets. We had no part in the onstruction of these datasets.

## 3.2 Compared methods

We compare our RBCC method against the following state-of-the-art and baseline E-MLC multi-label classification approaches:

*Recurrent Classifier Chain (RCC)* [15]: This method uses a unified RNN model to iteratively predict the classes that apply to each instance. The classes are ordered according to their frequency, with the most frequent classes being predicted before the rarest. The intuition is that the more frequent classes will be easier than the rarer classes, and thus the model will make fewer predictions early on if the frequent classes are predicted first. This would then mitigate cascading label errors. This class ordering is standard for classifier chains [19]. Note that there are three RCC variants proposed in [15]: $RNN^b$, $RNN^M$, and $Enc - Dec$. We refer to the $RNN^b$ approach as RCC, and is the method we compare against. We forego direct comparison with $RNN^M$, as it's main difference is that it predicts all positive classes first. This is likewise done by the extension of this method, $OF - RCC$, which we do compare against. Lastly, we do not compare against $Enc - Dec$ as it is specific to sequence data, which is not what we focus on in this work.

*Topological-Sorted Recurrent Classifier Chain (TS-RCC)* [15, 12]: This method likewise uses a unified RNN model to iteratively predict classes. However, before training the model a Bayesian network is fit to the class residuals, as is done for RBCC and is described in Section 2.2.1. Then, we perform a topological sort on the classes according the the Bayesian network, such that each class is placed after all of its parents. This ordering is then used for the class prediction of the RCC. While this method has not been proposed in the literature, it is the natural way of combining Bayesian networks with recurrent classifier chains.

*Order-Free Recurrent Classifier Chain (OF-RCC)* [4]: The OF-RCC uses the same base RNN architecture as the RCC and TS-RCC. However, instead of specifying an order for the classes to be predicted in, the OF-RCC *learns* the ordering itself. We compare against this method to validate that our RBCC's approach of predicting the classes in a principled order produces better results than relying on the model to learn an optimal ordering.

*Bayesian Classifier Chain (BCC)* [23]: Unlike the RCC models, the BCC utilizes an ensemble of individual models rather than a unified RNN classifier. Each class is given a corresponding classifier, such that the classifier takes in the observed features as well as the predictions for the *parents* of that class. The parents of each class are found by fitting a Bayesian network to the class residuals, as described in Section 2.2.1.

---

[1]http://host.robots.ox.ac.uk/pascal/VOC/voc2007/, https://www.flickr.com/help/terms
[2]http://www.uco.es/kdis/mllresources/#SceneDesc, license: PDDL
[3]http://www.uco.es/kdis/mllresources/#YelpDesc, license: `https://s3-media3.fl.yelpcdn.com/assets/srv0/engineering_pages/bea5c1e92bf3/assets/vendor/yelp-dataset-agreement.pdf`

| Evaluation Metrics | Methods | | | | | |
|---|---|---|---|---|---|---|
| | RBCC (Ours) | RCC | TS-RCC | OF-RCC | BCC | BD |
| Subset Accuracy ↑ | **0.240** ± 0.008 | 0.212 ± 0.002 | 0.192 ± 0.010 | 0.169 ± 0.009 | 0.210 ± 0.000 | 0.202 ± 0.002 |
| Hamming Loss ↓ | **0.186** ± 0.003 | 0.204 ± 0.001 | 0.209 ± 0.004 | 0.218 ± 0.004 | 0.199 ± 0.001 | 0.189 ± 0.000 |
| Macro-F1 ↑ | 0.556 ± 0.008 | 0.526 ± 0.004 | 0.506 ± 0.004 | **0.569** ± 0.004 | 0.551 ± 0.005 | 0.517 ± 0.008 |
| Micro-F1 ↑ | **0.670** ± 0.006 | 0.639 ± 0.002 | 0.628 ± 0.004 | 0.662 ± 0.004 | 0.653 ± 0.003 | 0.638 ± 0.003 |

Table 2: Results for the `Yelp` dataset. Bolded is the best performer, underlined is second best.

| Evaluation Metrics | Methods | | | | | |
|---|---|---|---|---|---|---|
| | RBCC (Ours) | RCC | TS-RCC | OF-RCC | BCC | BD |
| Subset Accuracy ↑ | **0.653** ± 0.007 | 0.548 ± 0.006 | 0.545 ± 0.007 | 0.610 ± 0.007 | 0.527 ± 0.008 | 0.479 ± 0.013 |
| Hamming Loss ↓ | **0.097** ± 0.002 | 0.116 ± 0.001 | 0.134 ± 0.002 | 0.102 ± 0.001 | 0.110 ± 0.002 | 0.105 ± 0.002 |
| Macro-F1 ↑ | 0.727 ± 0.004 | 0.610 ± 0.005 | 0.576 ± 0.010 | **0.742** ± 0.004 | 0.660 ± 0.009 | 0.650 ± 0.010 |
| Micro-F1 ↑ | 0.722 ± 0.005 | 0.650 ± 0.004 | 0.613 ± 0.005 | **0.735** ± 0.004 | 0.672 ± 0.006 | 0.652 ± 0.010 |

Table 3: Results for the `Scene` dataset. Bold is the best performer, underlined is second best.

*Binary Decomposition (BD)* [25, 8, 14]: Binary Decomposition is the process of transforming a multi-label classification problem into a set of binary classification problems. Specifically, a separate binary classifier is constructed per each class. Despite the fact that this method does not capture inter-class dependencies, it is a competitive approach for multi-label classification depending on the evaluation metric [8]. If performance is measured as a function of how many individual classes are predicted correctly, rather than if the label predicted label set is *exactly* correct, then BD is expected to do well.

Unlike the RCC models, the BCC and BD approaches utilize an ensemble of individual models rather than a unified RNN classifier. Each class is given a corresponding classifier such that the classifier takes in the observed features as well as the predictions for the *parents* of that class for the BCC, and only the observed features for BD. The parents of each class are found by fitting a Bayesian network to the class residuals, as described in Section 2.2.1.

## 3.3 Implementation Details

RBCC, RCC, TS-RCC, and OF-RCC all utilize the same RNN architecture (i.e., same number of layers and hidden sizes) and the same architecture for the classification layer. As the BCC and BD methods require an ensemble of models rather than a unified recurrent architecture, we replace the RNN with a feed-forward network for BCC and BD. However, the feed-forward network has a comparable number of parameters to the RNN and classifier used by the other methods in order to perform a fair comparison.

Experiments were performed on a computing cluster, using a Intel(R) Xeon(R) Platinum 263 8160 CPU @ 2.10GHz CPU, an NVIDIA Tesla V100 SXM2 GPU, and 128 GB of RAM.

Full details, including our specific architectural choices, are available in the Reproducibility Appendix. The code for RBCC is available at `https://github.com/waltergerych/RBCC`. More information is available in our supplemental materials.

## 3.4 Metrics

To evaluate the performance on each dataset, we use **four standard multi-label metrics**: *subset accuracy*, *Hamming Loss*, *macro F1*, and *micro F1*. Subset accuracy is the metric we explicitly optimize for as it directly corresponds to exact multi-label classification.

## 3.5 Classification Performance

We begin by demonstrating that our RBCC outperforms 5 state-of-the-art methods for multi-label classification. Results for the `Yelp`, `Scene`, and `PASCAL VOC 2007` datasets are shown in Tables 2, 3 and 4, respectively. Notably, our RBCC outperforms all other methods for each dataset for the

| Evaluation Metrics | Methods | | | | | |
|---|---|---|---|---|---|---|
| | RBCC (Ours) | RCC | TS-RCC | OF-RCC | BCC | BD |
| Subset Accuracy ↑ | **0.594** ± 0.009 | 0.457 ± 0.057 | 0.454 ± 0.008 | 0.575 ± 0.007 | 0.581 ± 0.011 | 0.578 ± 0.002 |
| Hamming Loss ↓ | **0.028** ± 0.000 | 0.045 ± 0.006 | 0.046 ± 0.000 | 0.030 ± 0.001 | 0.035 ± 0.001 | 0.029 ± 0.000 |
| Macro-F1 ↑ | 0.718 ± 0.016 | 0.495 ± 0.092 | 0.577 ± 0.011 | **0.756** ± 0.007 | 0.629 ± 0.001 | 0.746 ± 0.003 |
| Micro-F1 ↑ | **0.787** ± 0.006 | 0.646 ± 0.054 | 0.662 ± 0.006 | 0.782 ± 0.004 | 0.728 ± 0.005 | 0.781 ± 0.003 |

Table 4: Results for the `PASCAL VOC 2007` dataset. Bold is the best, underlined is second best.

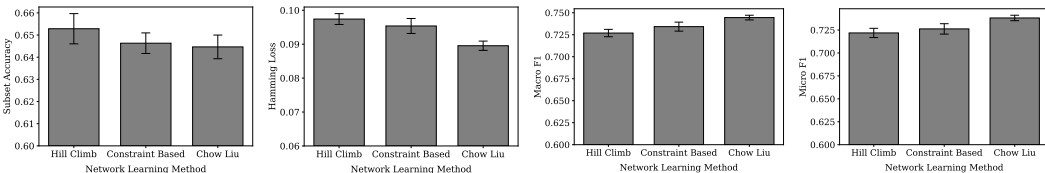

Figure 2: Observing the effect of the network learning algorithm on RBCC's performance. Results obtained from `Scene` dataset.

strict subset accuracy metric. This is important, as this metric corresponds to our problem definition (predicting the *exactly* correct label set).

The second-best method in terms of subset accuracy is usually one of the RCC methods, which is as expected as RCCs are the leading approach for maximizing subset accuracy. For the `PASCAL VOC 2007` dataset, the BCC performs second best, despite not utilizing parameter sharing during class prediction. This could be explained by `PASCAL VOC 2007` having more classes than the other datasets. Thus, the RCC approaches must perform inference over a longer sequence of classes, as they condition class prediction on all previously predicted classes. However, the BCC shares RBCC's approach of only conditioning the predictions for a given class on the parents of that class, resulting in the method *not* having to do inference over longer sequences even as the number of classes grows.

Our RBCC likewise performs the best on each dataset in terms of Hamming Loss. Unsurprisingly, binary decomposition performs second-best in terms of this metric. This is as expected, as Hamming Loss measures classification performance on a *per class, per instance* basis, rather than the label-set as a whole. Thus, it is reasonable that the binary decomposition approach of maximizing the probability for each class individually results in good performance.

Lastly, RBCC performs well on the Micro-F1 metric (the best on two datasets, and second-best on the third), but is second-best in terms of Macro-F1 for two datasets and third-best for one. This discrepancy in f1-scores is likely explained by Macro-F1 being more sensitive to class imbalance. Even if a class appears very infrequently, it is weighted equally to more frequent classes. This is in contrast to our primary target of subset accuracy.

### 3.6 Effect of Network Learning Algorithm

Using the Inter-class Dependency Network component of RBCC, we fit a Bayesian network of class inter-dependencies to the observed data. There exist several choices for the learning algorithm used to fit the Bayesian network. In this experiment, we study the effect of the choice of learning algorithm on RBCC's classification performance. Specifically, we test the performance with three standard Bayesian network approaches: Hill Climbing [22, 6], Constraint Based [22, 27], and the Chow Liu algorithm [5].

The Hill Climbing approach is a greedy local search [22]. Starting with a disconnect DAG, single-edges are added or removed in order to improve the likelihood score of the model. Once a local optimum is obtained, the search ceases.

The Constraint Based approach [22, 27] identifies in-dependencies in the data using a Chi-Squared hypothesis test. The Bayesian network is then constructed according to this independence test.

The Chow-Liu algorithm finds the maximum-likelihood graph such that each node (class) has at most one parent. First, the mutual information for all class pairs is calculated. Next, the maximum weight spanning tree is found according to the obtained mutual information. Lastly, a random node is picked

to be the root node, with directions assigned to each edge radiating outwards from the root node, obtaining a directed graph from the undirected maximum weight spanning tree.

The results of this experiment on the `Scene` dataset are shown in Figure 2. Importantly, the performance of RBCC is very similar for each network learning approach, as the performance difference between methods is typically within the margin of error of each other. This indicates that RBCC is robust to the choice of this algorithm.

There is a slight performance increase (roughly 0.5%) for the Hill Climbing approach in terms of Subset Accuracy, indicating that it might be the more appropriate choice as we are aiming to maximize this metric in particular. Indeed, the Hill Climbing approach is the default for RBCC, as described in Section 2.2.1.

Interestingly, the Chow Liu algorithm outperforms the other approaches by a small margin in terms of the other metrics (Hamming Loss, Macro-F1, and Micro-F1), despite performing the worst for subset accuracy, it being the more restrictive network learning approach. This might be explained by the fact that the Chow Liu algorithm assigns fewer parents to each class on average than the other approaches (with an average of 0.86 parents per class for Chow Liu and 1.67 for Hill Climbing). Thus, this approach might learn less-accurate inter-class dependencies, explaining why it gives the lowest subset accuracy score. However, for the metrics where capturing inter-class dependencies is less important, having fewer parents results in shorter sequences to perform inference over and may lead to better predictions on individual classes rather than label sets.

## 4    Limitations

RBCCs faces the same limitation that nearly every exact classification suffers from: Predicting the label set *exactly* correct becomes exponentially difficult as the size of the set of possible labels grows. Thus, performance will degrade in settings where there are very many possible labels.

**Negative societal impact.** Our method improves over standard recurrent classier chains, which are commonly used in many object detection and image recognition systems. Image recognition systems that can recognize individuals or members of certain minority groups have recently raised ethical concerns [26]. However, such systems are not the focus of this work, and RBCCs are not any more prone to ethical misuse than are any other classifier chain or multi-label classification approaches.

## 5    Conclusion

In this paper we propose RBCC, an effective method for exact multi-label classification that improves on recurrent classifier chains' ability to model inter-class dependencies using principled class ordering and by conditioning the predictions for each class on only its parents. Our principled class ordering and the parent classes themselves are found using by fitting a Bayesian network to the classes. Through extensive experiments we show that RBCC consistently outperforms the leading methods for exact multi-label classification. In particular, RBCC outperforms the other methods in all cases with respect to the strict Subset Accuracy metric, which is the metric that directly measures exact classification. Additionally, we find that RBCC is robust to the choice of network learning method. By showing the benefit of combining recurrent classifier chains with Bayesian networks, which has to-date been unstudied, we believe our results can inspire additional research on combining class network models with recurrent classifier chains.

## 6    Acknowledgements

This material is based on research sponsored by DARPA under agreement number FA8750-18-2-0077. The U.S. Government is authorized to reproduce and distribute reprints for Governmental purposes notwithstanding any copyright notation thereon. The views and conclusions contained herein are those of the authors and should not be interpreted as necessarily representing the official policies or endorsements, either expressed or implied, of DARPA or the U.S. Government. Additionally, we would like thank the WASH Research Group at WPI as well as the DAISY Lab at WPI for the invaluable feedback during the research process.

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
