# Appendix For
# Recurrent Bayesian Classifier Chains For Exact Multi-Label Classification

**Walter Gerych, Thomas Hartvigsen, Luke Buquicchio, Emmanuel Agu, Elke Rundensteiner**
Worcester Polytechnic Institute
Worcester, MA
{wgerych, twhartvigsen, ljbuquicchio, emmanuel, rundenst}@wpi.edu

## 1 Implementation Details

**Base RCC Architecture** The recurrent methods (RBCC, TS-RCC[6, 3], RCC [6], and OF-RCC [1]) were each implemented in PyTorch [7]. Each method consisted of a 1-layer GRU with a hidden space size of 100. Additionally, each recurrent method had a 1-layer feed forward network to map from the 100 dimensional latent space into prediction probabilities.

**Base Feed Forward Network Architecture** The non-recurrent methods (BCC and BD) consisted of an ensemble of feed-forward networks that each map from the feature space to a 100 dimensional latent space, replacing the GRU of the recurrent methods. These methods likewise had an additional feed-forward layer to map from the latent space into prediction probabilities. The networks were likewise implemented in PyTorch [7].

**Bayesian Network Learning** We use the `bnlearn` [8] Python package for Bayesian network learning. For the experiments described in Section 3.5 of the main paper, all methods which required a Bayesian network (RBCC, TS-RCC, BCC) used the same network which was found using the *Hill Climbing* method in `bnlearn`. For the experiment described in Section 3.6, we used the *Constraint Based*, *Hill Climbing*, and *Chow Liu* algorithms in `bnlearn`.

**Obtaining Class Residuals** We fit the Bayesian network on class *residuals* rather than on the classes themselves. These residuals are obtained by first training a separate classifier per each class, and then calculating the residual as the error between the predicted and ground truth class. These classifiers all had the same form as the base model used by the BD method, described above. As calculating these residuals requires out-of-sample inference, we fit the models and half of the data and evaluate on the other half, before switching the training and testing sets and training/inferring again. Thus, the residual of each class for each point is obtained from a model that was *not* trained on that point.

**Training Hyperparameters** For each method, we used a batch size of 128 and a learning rate of 0.001. We used the Adam optimizer [4] and PyTorch's exponential learning rate scheduler with gamma set to 0.99. Each method was trained until convergence for 200 epochs.

**Feature Representations** The `Yelp` and `Scene` datasets were already pre-featurized, but the `PASCAL VOC 2007` dataset comes in the form of raw images. We thus featurize the `PASCAL VOC` using a pre-trained network. Specifically, we used the pretrained ResNet-18 model available in PyTorch, and extracted the feature representations from the final average pooling layer.

## 2 Importance of Non-Noisy Conditioning

Rather than conditioning the prediction for each class on all previously predicted classes, as is done by RCCs, RBCC instead conditions the prediction for each class on *only* the parents of that class. We posit that conditioning on all previously-predicted classes introduces noise into the class prediction

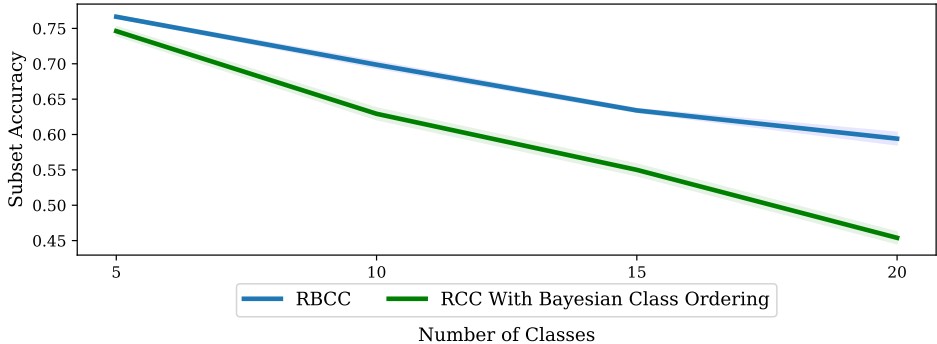

Figure 1: RBCC, which conditions the predictions for each class only on its parents, outperforms a comparable RCC which conditions each class on its parents *and all previously predicted classes*. The disparity in performance increases as the number of classes grows. Results shown for the `PASCAL VOC 2007 dataset.`

Table 1: Performance of each method on the Enron dataset.

| Metric | RBCC (Ours) | RCC | TS-RCC | OF-RCC | BCC | BD |
|---|---|---|---|---|---|---|
| Subset Acc. | **0.305**+/-0.016 | 0.100+/-0.001 | 0.090+/-0.005 | 0.1470.019 | 0.090+/-0.007 | 0.2570.020 |
| Hamming Loss | **0.027**+/-0.004 | 0.043+/-0.003 | 0.041+/-0.001 | 0.040+/-0.003 | 0.041+/-0.002 | 0.030+/-0.001 |
| Macro-F1 | **0.233**+/-0.013 | 0.0051 +/- 0.051 | 0.054+/-0.001 | 0.079+/-0.006 | 0.052+/-0.003 | 0.199+/-0.018 |
| Micro-F1 | **0.759**+/-0.075 | 0.607+/-0.020 | 0.614+/-0.023 | 0.617+/-0.022 | 0.611+/-0.022 | 0.716+/-0.012 |

for the classes that are independent of each other. This leads to cascading label errors having more negative effects on the final label set. To validate that our "non-noisy" class conditioning approach is indeed better than the approach of standard RCC's, we perform an additional experiment.

Here, we compare the performance of RBCC against a RCC that predicts the classes in an order determined by a Bayesian network. Importantly, the Bayesian network is the same as the one used by RBCC, and the class ordering implies that each class is predicted before its parent classes. However, the RCC conditions each class on *all* previously predicted classes. The two approaches are compared on the `PASCAL VOC 2007` dataset where we vary the number of classes from 5 to 20 (the maximum number of classes in `PASCAL VOC 2007`), such that the classes chosen are the most frequent.

Results shown in Figure 1. As expected, RBCC's approach of leveraging conditional independence by performing "non-noisy" conditioning results in improved classification performance. Notably, the difference in performance increases as the number of classes increases. This fits our hypothesis that conditioning on all previously predicted classes negatively impacts performance, as the sequence length over which inference is performed grows as the number of classes for the RCC. For the RBCC, on the other hand, the sequence over which inference is performed is only as long as the number of parents, which will nearly always be significantly less than the total number of classes.

## 3   Additional experiments

We expand our experimental evaluation of our method to 3 additional datasets of varying numbers of classes: Enron (53 classes) [5], EukaryoteGO (22 classes) [9], and Yeast (14 classes) [2]. Results (shown in Tables 1-3) indicate that our method continues to consistently outperform the state-of-the-art methods.

Table 2: Performance of each method on the EukaryoteGO dataset.

| Metric | RBCC (Ours) | RCC | TS-RCC | OF-RCC | BCC | BD |
|---|---|---|---|---|---|---|
| Subset Acc. | **0.906**+/-0.041 | 0.813+/-0.018 | 0.792+/-0.020 | 0.863+/-0.008 | 0.771+/-0.029 | 0.766+/-0.007 |
| Hamming Loss | **0.006**+/-0.003 | 0.011+/-0.007 | 0.010+/-0.005 | 0.008+/0.001 | 0.008+/-0.002 | 0.013+/-0.000 |
| Macro-F1 | **0.907**+/-0.053 | 0.600+/-0.009 | 0.590+/-0.023 | 0.629+/-0.014 | 0.537+/-0.020 | 0.466+/-0.028 |
| Micro-F1 | **0.937**+/-0.028 | 0.892+/-0.024 | 0.863+/-0.030 | 0.921+/-0.005 | 0.843+/-0.025 | 0.872+/-0.006 |

Table 3: Performance of each method on the Yeast dataset.

| Metric | RBCC (Ours) | RCC | TS-RCC | OF-RCC | BCC | BD |
|--------|-------------|-----|--------|--------|-----|-----|
| Subset Acc. | **0.243**+/-0.016 | 0.150+/-0.008 | 0.148+/-0.011 | 0.219+/-0.022 | 0.153+/-0.013 | 0.121+/-0.014 |
| Hamming Loss | 0.195+/-0.008 | 0.254+/-0.006 | 0.257+/-0.005 | **0.188**+/-0.005 | 0.248+/-0.010 | 0.203+/-0.008 |
| Macro-F1 | **0.420**+/-0.015 | 0.282+/-0.053 | 0.279+/-0.050 | 0.394+/-0.010 | 0.291+/-0.055 | 0.293+/-0.012 |
| Micro-F1 | 0.658+/-0.025 | 0.528+/-0.021 | 0.530+/-0.019 | **0.682**+/-0.012 | 0.526+/-0.024 | 0.602+/-0.022 |

# 4 RBCC Algorithms

---

**Algorithm 1:** Algorithm to obtain Bayesian network of conditional inter-class dependencies

---

function get_conditional_network $(g(\cdot), \mathbb{D}_{train}, S)$;

**Input**      : Binary relevance classifier $g(\cdot)$, training dataset $\mathbb{D}_{train} = \{X_{train}, Y_{train}\}$,
             directed acyclic graph fitting scheme $S$

**Output**    : Bayesian network of inter-class
             dependencies $\mathcal{G}$

\\As we desire out-of-sample inference when determining residuals, we split train set into K
  subsets

**for** $k \leftarrow 1$ **to** $K$ **do**
  | $\mathbb{D}_k = n/k$ points sampled without replacement from $\mathbb{D}_{train}$
**end**

$all\_residuals = \{\}$

**for** $k \leftarrow 1$ **to** $K$ **do**
  | $g.fit(X_k, Y_k)$
  | $\{E_1, E_2, \ldots, E_L\}_k = Y_{train} - g.predict(X_{train} \setminus X_k)$
  | $all\_residuals.concatenate(\{E_1, E_2, \ldots, E_L\}_k)$
**end**

$\mathcal{G} = S.fit\_predict(all\_residuals)$

return $\mathcal{G}$

---

---

**Algorithm 2:** Training algorithm for RBCC

---

function train_RBCC $(\mathbb{D}, \mathcal{G})$;

**Input** : Training dataset $\mathbb{D}$, Bayesian network $\mathcal{G}$

**Output** : Trained Recurrent Conditional Dependency Model $r_\Phi$ and Bayesian-Conditioning
  Classifier classifier $h_\Phi$

**for** $epoch \leftarrow 1$ $to$ $num\_epochs$ **do**
  **for** $batch$ $in$ $batches$ **do**
    **for** $\boldsymbol{x}_b, \boldsymbol{y}_b$ $in$ $batch$ **do**
      **for** $k = 1$ $to$ $L$ **do**
        $\boldsymbol{a}_k = initialize\_hidden\_state$
        **if** $Pa(C_k) = \emptyset$ **then**
          $\boldsymbol{v}_b = <\boldsymbol{x}_b, \boldsymbol{0}^L>$
          $\boldsymbol{a}_k = r_\Phi(\boldsymbol{v}_b, \boldsymbol{a}_k)$
          $h\_b\_k = h_\Phi(\boldsymbol{a}_k)$
        **else**
          **for** $C_\ell$ $in$ $Pa(C_k)$ **do**
            $\boldsymbol{v}_b = <\boldsymbol{x}_b, [\boldsymbol{y}_b]_\ell>$
            $\boldsymbol{a}_k = r_\Phi(\boldsymbol{v}_b, \boldsymbol{a}_k)$
          **end**
          $h\_b\_k = h_\Phi(\boldsymbol{a}_k)$
        **end**
      **end**
    **end**
    Calculate loss according to Equation 4 (in the main paper)
    Update $\Phi$ according gradient descent
  **end**
**end**
return $r_\Phi, h_\Phi$

---

---

**Algorithm 3:** Inference algorithm for RBCC

---

function RBCC_predict_class$(T, \mathcal{G}, r_\Phi, h_\Phi, k, x)$;

**Input** : Hash table of previously predicted classes $T$, Bayesian network $\mathcal{G}$, trained RBCC
  RNN and feed forward network $r_\Phi$ and $h_\Phi$

**Output** : $\boldsymbol{P}(C_k|Pa(C_k), x)$

**for** $C_\ell$ $in$ $Pa(C_k)$ **do**
  **if** $if$ $c_\ell$ $not$ $in$ $T$ **then**
    $T[\ell] = round(predict\_class(T, \mathcal{G}, r_\Phi, g_\Phi, \ell, x))$
  **end**
**end**
$\boldsymbol{a}_k = initialize\_hidden\_state$
**if** $Pa(C_k) = \emptyset$ **then**
  $V = <x, \boldsymbol{0}^L>$
  $p\_k\_x = h_\Phi(r_\Phi(V, \boldsymbol{a}_k))$
  $T(c_k) = round(p\_k\_x)$
**else**
  **for** $C_\ell$ $in$ $Pa(C_k)$ **do**
    $\boldsymbol{v} = <x, T(c_\ell)>$
    $\boldsymbol{a}_k = r_\Phi(V, \boldsymbol{a}_k)$
  **end**
  $p\_k\_x = h_\Phi(\boldsymbol{a}_k)$
  $T(c_k) = round(p\_k\_x)$
**end**
return $p\_k\_x$

---