# OpenReview forum: "Recurrent Bayesian Classifier Chains for Exact Multi-Label Classification"
_NeurIPS.cc/2021/Conference — NeurIPS 2021 Poster_

### Official Review · Reviewer_RLfS · 2021-07-14

**Rating:** 7
**Confidence:** 4

**Summary:**

This paper proposes a new method, Recurrent Bayesian Classifier Chains (RBCC), for exact multi-label classification problems (maximizing subset accuracy). RBCC involves three steps: 1. Intra-class Dependency Network to learn a Bayes net to the classes; 2. Recurrent Conditional Dependency Model to summarize the information of features and the corresponding parent classes learned in the previous step; 3. Bayesian-Conditioning Classifier to predict. RBCC is shown to outperform several state-of-the-art (STOA) algorithms on some real-world multi-label data sets with respect to subset accuracy.

**Ethical Concerns:**

No ethical issues.

**Limitations And Societal Impact:**

Yes.

**Main Review:**

Overall I think the method (RBCC) proposed in this paper is relatively novel and its performance is good. The description of the method is fairly clear, and enough background is provided. My main concern is about the experiments (see below). The following are my comments on each of the dimension.

Originality.
The method (RBCC) is new. Although each component is known, the combination is relatively novel, for example, how to better incorporate a Bayes net into the recurrent classifier chain system. The related work is adequate and some STOA algorithms are described and compared.

Quality.
The work is sound. The methods for designing RBCC are appropriate. However, there are some issues with the experimental design, which I list below.

1) Why some data sets (Reuters-21578, RCV1-v2 and BioASQ) in paper [15] are not used in the experiments? Both paper [15] and this paper aim to maximize subset accuracy, and methods in both papers are based on Recurrent Classifier Chain (RCC), so it is reasonably expected the authors should use one of the data sets from [15] or at least provide explanations for not using.

2) I might be missing something, but I could not find the number of repeated experiments in the paper, given that the error bars are reported.

3) Training time and inference time are not discussed. It looks that since RBCC has slightly more complicated structures than RCCs, it might take longer to train. Unfortunately this info is lack in the paper. The authors should discuss the computational costs.

Clarity.
Overall this paper is clear. Enough background is provided. The descriptions of the proposed method is detailed and clear. Nevertheless, some terms are not clearly defined and there are several typos (some of which are listed below). If this paper is accepted, the authors should proofread carefully, add necessary explanations and fix typos. Question regarding some terms.

1)  What is "non-noisy"? I can get a sense of the meaning after reading the whole paper, but I could not find where this is defined.

2) In the contributions, it is stated that a novel training algorithm is proposed. Which part of the paper does this "training algorithm" refer to?

Significance.
The method (RBCC) proposed in this work is useful for researchers and practitioners working on multi-label learning. It also advances STOA in exact multi-label classification problems.

---
Extra question (the authors do not need to respond):

1) Could the authors also report metric F1 (instance-averaged) in the experiments?
---
Typos:

Page 3, line 100, duplicate citations [22]

Page 3, line 113, "strongly dependent class" -> "strongly dependent on class"

Page 3, line 122, $P(C_k | x) \ne P(C_k | C_n, x)$ -> $P(C_k | x) = P(C_k | C_n, x)$

Page 5, line 187, $V = $ -> $V_t =$

Page 6, line 243, "in the onstruction" -> "in the construction"

---

After rebuttal: My main concerns (1 and 3 in the Quality section) are addressed. I keep my original rating for this paper.

---
Citations:
[15] Jinseok Nam, Eneldo Loza Mencía, Hyunwoo J Kim, and Johannes Fürnkranz. 2017. Maximizing subset accuracy with recurrent neural networks in multi-label classification. In Advances in Neural Information Processing Systems. 5413–5423.


**Time Spent Reviewing:**

8 hours

---

> ### Author Response · Authors · 2021-08-10
> **Thank you for your thoughtful review**
>
> Thankful for your constructive review. We are happy to see that you found the method to be novel and the paper to be clear. We address your specific comments below.
>
>
> **”Why some data sets (Reuters-21578, RCV1-v2 and BioASQ) in paper [15] are not used in the experiments?”**
>
> These datasets are characteristic of Extreme Multi-Label Classification (EMLC) as they have very large label sets, a setting in which subset accuracy is inappropriate. Indeed, this is one of the few issues of [15] that the NeurIPS reviewers appeared to have, based on their publicly available reviews. To quote one of those reviewers for the original use of these datasets: **“It is also a bit strange to optimize subset zero-one loss in MLC applications with hundreds of labels. Subset zero-one loss is for sure not the measure of interest in extreme MLC applications”**. For this reason we choose more standard multi-label datasets for which optimizing subset accuracy is appropriate.
>
> **”I could not find the number of repeated experiments in the paper, given that the error bars are reported.”**
>
> Thank you for pointing this out. The experiments were run 10 times each. We will explicitly state this in the camera-ready version of the paper.
>
> **”It looks that since RBCC has slightly more complicated structures than RCCs, it might take longer to train.”**
>
> Our RBCC requires fitting a Bayesian network of the label dependencies prior to training. However, this is a one-time cost that does not need to be repeated during training, or when doing hyperparameter tuning. This cost of fitting the network varies depending on the algorithm used to find the network. A tradeoff between goodness of fit and inference time can be made for the network finding algorithm, but the specific method used to find the dependency network is not the focus of our work.
>
> Once the dependency network is obtained, training of our RBCC does not vary much in complexity from standard RCCs. As we read in the parents of each class separately when making class predictions, at worst our recurrent network must make N steps for every 1 step of the RCC, where N is the number of parents of the class with the most parents. This is not bad, as this is just a constant scaling of the running time of a comparable RCC. Additionally, in reality the network will usually make less than N steps, as classes will have less than the maximum number of parents.
>
> **”What is "non-noisy"? I can get a sense of the meaning after reading the whole paper, but I could not find where this is defined.”**
> We refer to not conditioning class inference on uncorrelated classes as “non-noisy” class conditioning. Thank you for alerting us that this is not clearly defined in the original version of the paper. We have remedied this for the camera-ready version.
>
> **”In the contributions, it is stated that a novel training algorithm is proposed. Which part of the paper does this ‘training algorithm’ refer to?”**
>
> This process is described in lines 209 to 222. In the camera-ready version, we will make use of the additional allowed page of content to add in pseudocode for the training algorithm.
>
> --------
>
> Lastly, thank you for the typos you have identified. We have remedied them.
>
> -----------
>
> References
>
> Paper 2802 Reviews (Maximizing Subset Accuracy with Recurrent Neural Networks): https://proceedings.neurips.cc/paper/2017/file/2eb5657d37f474e4c4cf01e4882b8962-Reviews.html

---

### Official Review · Reviewer_JRCf · 2021-07-15

**Rating:** 6
**Confidence:** 4

**Summary:**

This paper presents a new method for optimizing the subset zero-one loss in multi-label classification. The new method is an adaptation of probabilistic classifier chains. The central idea is to replace the chain by a dependence structure learned using Bayesian networks. In theory the order of the labels in a probabilistic classifier chain does not matter, but in practice one can observe performance drops if the chosen order leads to propagation of errors. By replacing a random order with the dependency structure of a Bayesian neural network, the authors intend to solve the problem of choosing an appropriate order.

As underlying models to fit probabilities, recurrent neural networks are used. Experimental results are reported for three benchmark datasets.

**Ethical Concerns:**

No concerns.

**Limitations And Societal Impact:**

They have a paragraph about this in the paper. No concerns from my side.

**Main Review:**

Overall this is a nicely-written paper that presents a novel idea. Indeed, it is the case that in practice the order of labels in a probabilistic classifier chain matters. Learning the dependency structure by means of a Bayesian network is therefore a useful thing to do. Intuitively, I have the feeling that the presented approach makes sense. On the other hand, the presented approach is not very novel, since in reference [30] something similar was done, albeit without a focus on subset zero-one loss and without recurrent neural networks as underlying models. However, I do believe there is enough novelty to justify a new paper on this topic.

The existing literature is covered quite well, and in the experiments the choice of baselines is appropriate to my opinion. With only three datasets, the number of datasets in the experiments is quite small, especially because there are much more datasets in the public domain, see e.g. the MULAN datasets. I believe that it is important to justify why only those three datasets were analyzed in order to avoid being accused of cherry picking. I can imagine that those datasets were chosen because they have more complex feature spaces than the simple MULAN datasets, but it would be good to state that explicitly.

At some points the mathematical notation is a bit sloppy. In general it is important to formally define all notations, while being consistent in the use of vectors, components of vectors, sets and random variables. An example of an awkward notation is line 159, where C_ki = g_k(x_i) + E_ki. The model g is here a classifier that probably outputs zero or one, but C_ki is a pointer to class k. It might be better to use the notation Y_i, as defined on page 2, and refer to the k-th entry in Y_i. Also in Lemma 1, the quantities E_1,...,E_L are used but they are not formally defined. Most likely they allude to the vectors that originate by grouping E_i,k for all i. Another example of an incorrect notation is Eqn. 1, because the expectation over X is not needed there. If you want to refer to parameter fitting, please parameterize the distribution P by theta and formulate the optimization problem as the minimization of the log loss over a training dataset. Eqn. 1 looks very awkward for the moment.

I don't find the title very appropriate, and I would suggest to change it. When I downloaded the pdf, I thought the paper would be about a Bayesian variant of probabilistic classifier chains, in which a distribution of models is learned instead of a single model, similar to Bayesian methods such as Gaussian processes and Bayesian neural networks. However, the term Bayesian here comes from the fact that Bayesian networks are used to learn dependencies between labels. For example, a title like "Learning class dependencies for probabilistic classifier chains" or something similar would be more appropriate.


**Time Spent Reviewing:**

3

---

> ### Author Response · Authors · 2021-08-10
> **Thank you for your thoughtful review**
>
> We thank you for your thoughtful and constructive review. We are very pleased to see that you found our work to be ”a nicely-written paper that presents a novel idea.” We respond to your specific comments below.
>
> **====Choice of datasets====**
>
> You are correct in your assessment that our datasets were chosen as they have complex feature spaces (as they correspond to complex, real-world tasks). We will state this more explicitly in the camera-ready paper. Additionally, we have now added additional experiments on several MULAN datasets. Specifically, we compare our method against the state-of-the-art methods on the Enron , EukaryoteGO, and Yeast datasets. These were chosen as they have varying numbers of classes (53, 22 and 14, respectively). We continue to outperform the state-of-the-art methods in all three additional datasets. Please see Tables 1-3 in our General Response post for the detailed result table.
>
> **====Notation====**
>
> Thank you for the detailed suggestions on how to improve clarity of our presentation. We have integrated your suggestions into the paper. For instance, the undefined quantities in Lemma 1 are now given a separate definition, and Equation 1 is now expressed as an argmax of the parameter $\theta$. We unfortunately cannot upload a revised version of the paper during this particular author response period, but the changes were straightforward.
>
> **====Title of the paper====**
>
> Thank you for this suggestion. We are fine with your suggestion to change the title of the work to “Learning class dependencies for probabilistic classifier chains” for the camera-ready version in order to not draw unintended parallels to Bayesian neural networks.

---

### Official Review · Reviewer_DcQS · 2021-07-16

**Rating:** 5
**Confidence:** 4

**Summary:**

The authors propose Recurrent Bayesian Classifier Chains (RBCCs), a new method for exact multi-label classification. It combines the idea of ordering and conditioning labels based on a Bayesian network, utilized previously in Bayesian Classifier Chains (BCCs), and using a recurrent neural network as in Recurrent Classifier Chain (RCC). Usage of recurrent architecture allows modeling that condition on previously predicted classes instead of using many separate binary models originally used in the Classifiers Chain. The proposed approach uses an established approach to construct a Bayesian network. The novelty of the approach lies in the training and inference procedure. Instead of performing many to many classification, where labels are predicted one after another as in the case of standard RCCs, in RBCCs, many to one classification is performed for each class. Each class is conditioned only on features and labels it depends on in the obtained Bayesian network. The author validates the attractiveness of their method in empirical study on three popular benchmark datasets with a small number of labels. In all three cases, the proposed approach significantly improves in terms of subset accuracy over the baselines.

**Ethical Concerns:**

No ethical issues were found.

**Limitations And Societal Impact:**

The limitations were addressed, no issues with negative social impact were found.

**Main Review:**

Strengths:
+ ~~The organization of this paper is good, and it's easy to read and understand.~~
+ The proposed method seems to be an intuitive and neat improvement to BCCs/RCCs formula.
+ The RBCCs consistently outperform the benchmark in the empirical evaluation.
+ Comparison of different Bayesian network construction technique and their impact on the predictive performance of RBCs.
+ Comprehensive description of implementation details in the appendix.

Weaknesses:
- Empirical evaluation was conducted on only three datasets, all with a small number of labels - this is the only clear issue I have. With so many multi-label datasets from varying domains and a varying number of labels. I believe that a much more extensive empirical evaluation should be performed.

Other comments:
* The simplest baseline that directly optimizes argmax E[P(Y|x)] is the well-known labels power set method. Why not include it as a simple baseline, especially when comparing datasets with small labels?
* "If the parents have not yet been predicted, we recursively call the inference algorithm on the unpredicted parent classes." - this is not necessary since the Bayesian network is a DAG, you just need to predict classes in their topological order.
* There should be an equal sign instead of unequal in line 122.

---

Edit after the Authors' response: Since the authors addressed my main issue by conducting additional experiments, I'm raising my rating (to 6).

---

Edit after further discussion with AC and the Reviewers: Further discussion highlighted several issues with notation and clarity of the paper that led to differences in understanding of the proposed method among the Reviewers. After consideration, I think the paper requires major revision, and despite attractive results, should not be accepted to the conference. Because of that, I'm reverting my rating to the original one (5).

**Time Spent Reviewing:**

6

---

> ### Author Response · Authors · 2021-08-10
> **Thank you for your thoughtful review**
>
> Thank you for your thoughtful review. We are very pleased to see that the fact that “empirical evaluation was conducted on only three datasets, all with a small number of labels - this is the only clear issue” you have. We respond to this by adding in 3 additional datasets to our analysis, doubling the number of datasets we evaluate on. We use the Enron, EukaryoteGO, and Yeast datasets (all standard multi-label benchmarks). We continue to outperform the state-of-the-art in all three additional datasets. We kindly refer you to Tables 1-3 in our General Response post for the specific results.

---

> > ### Comment · Reviewer_DcQS · 2021-08-18
> > **Rasing my rating**
> >
> > Dear authors,
> >
> > thank you for providing additional experiments and answering other reviewers' questions. Since new results confirm the superiority of your method, I would like to raise my score to 6.
> >
> > Additional comment: It seems that the advantage of your method on the Erlon dataset is much more significant. This makes sense, since in many cases, when a number of labels are larger, many of them will not be correlated. This should result in a sparse Bayesian network, possibly resulting in shorter labels' sequences and making the training faster and easier than in the baselines. Maybe as a feature work, it's a good idea to investigate the performance on even a little bit larger datasets and the sparsity of the Bayesian network.

---

> > > ### Author Response · Authors · 2021-08-19
> > > **Thank you for raising your score**
> > >
> > > We are happy that you agree that we have demonstrated a decisive win for our method by incorporating additional datasets, and we greatly appreciate that you have raised your score.

---

> > > > ### Comment · Reviewer_DcQS · 2021-09-16
> > > > **Reverting to my original score**
> > > >
> > > > Dear Authors,
> > > >
> > > > The further discussion with AC and the other Reviewers highlighted several issues with notation and clarity of your paper that apparently led to differences in understanding of the details of the proposed method among the Reviewers. After rereading your work, I agree that it requires major revision. Unfortunately, the number of necessary changes and their scopes are both large. Because of that, I cannot recommend accepting your work, and I reverted my rating to the original one.

---

> > > > > ### Author Response · Authors · 2021-09-20
> > > > > **We respectfully but strongly disagree**
> > > > >
> > > > > We are disappointed to hear that you are lowering your score, and respectfully but strongly disagree with your decision. We hope you have time to read below and reconsider.
> > > > >
> > > > > The AC's comments are primarily discussion of notation, and we have already outlined how we will change the notation to a more standard style (though our original notation was internally consistent).
> > > > >
> > > > > Beyond notation, we have provided the detailed verbiage of clarifications the reviewers and the AC asked for (in our responses in the review system), which are:
> > > > >
> > > > > * Tweak text around Equation 3
> > > > > * Expand related work discussion on differences between our approach and 3 RCC variants [15].
> > > > >
> > > > > These changes consist of a simple replacement and minor clarifications (above) written during the discussion with the AC. In fact, we have the revised manuscript ready, and it took far less than a single day.
> > > > >
> > > > > Beyond that, no actual flaw of our work has been pointed out by the AC. Plus, the reviewers have already stated that our work is novel, useful, and a clear improvement over the state-of-the-art Recurrent Classifier Chains and Bayesian Classifier Chains.
> > > > >
> > > > > Also, you indicated in your initial review that the organization of the manuscript was good and easy to understand. During the rebuttal you said that your concerns were addressed. We are thus baffled and saddened that you are contemplating lowering your score. We are concerned that such a score change may affect the final accept decision. Please please please put yourself into our shoes.

---

### Official Review · Reviewer_Bp9h · 2021-07-17

**Rating:** 6
**Confidence:** 4

**Summary:**

This paper proposed Recurrent Bayesian Classifier Chains (RBCC) for exact multi-label classification. The proposed method learned a Bayesian network of class dependencies to improve the Recurrent Classifier Chains (RCCs). The comparative experiments were performed with RCC ([15], 2017, Nam, etc.), TS-RCC ([15], 2017, Nam, etc.), OF-RCC ([4], 2018 , Chen, etc.), and BCC ([23], 2014, Sucar). And the results demonstrated the effectiveness of the RBCC method.

**Limitations And Societal Impact:**

## Weakness:

- In equation 1:
The left side of equation is a function of x, while the right side is the expectation on x, that means the right side is irrelevant to x.

- line 16, ‘we outperform the state of the art methods’
More experiments should be conducted to compare with some latest methods proposed in 2019-2020.

- page 5-6, Table 1-3
The improvement of this method is marginal.

## Minor issues:
	- line 100:
	The reference [22] needs to be mentioned only once.

## Suggestion:
- Some figures could be demonstrated to help the reader to understand the method and the advantage of the method, such as, the algorithm flow chart and the resulting figure which can illustrate the superiority of the proposed algorithm.



**Main Review:**

- The RBCC incorporates a Bayesian network into the recurrent classifier chain system.
- A new algorithm is proposed for training and inference.
- Experiments show that this method improves the performance of the mentioned methods.


**Time Spent Reviewing:**

6 hours

---

> ### Author Response · Authors · 2021-08-10
> **Thank you for your thoughtful review**
>
> We thank you for your thoughtful review. We respond to your specific comments below.
>
> **”The left side of equation [1] is a function of x, while the right side is the expectation on x, that means the right side is irrelevant to x.”**
>
> Thank you for identifying this typo. We have correctly reformulated the equation to be an argmax of the parameters $\theta$.
>
> **”More experiments should be conducted to compare with some latest methods proposed in 2019-2020.”**
>
> The purpose of our work is to show the benefit of incorporating Bayesian networks explicitly into the training of Recurrent Classifier Chains, and to validate this we compare against all fundamental classifier chain approaches (classic Classifier Chains, Recurrent Classifier Chains, Recurrent Classifier Chains with topological sorting, and Bayesian Classifier Chains). While there have been classifier chain papers in the past year that propose specific architectures designed for different domain use cases, that is not the focus of this work. We believe we have identified and compared against all fundamental classifier chain variants. If there is a specific method that should be included in our study, we are happy to include a comparison against it in our camera ready version.
>
> **”Table 1-3 The improvement of this method is marginal”**
>
> The performance of our method is statistically significantly better than the state-of-the-art on all datasets for the Subset Accuracy metric, which is our target metric and the focus of this work. Our method is also statistically significantly better for the remaining metrics in most other cases.
>
> **”Some figures could be demonstrated to help the reader to understand the method and the advantage of the method”**
>
> Thank you for this suggestion. We have such a figure created already, and only left it out due to space constraints. We will include it in the camera-ready version, making use of the additional page allowance.

---

> ### Comment · Reviewer_Bp9h · 2021-08-26
> **raise my score to 6**
>
> Thanks for the rebuttals from the authors.
>
> After reading the rebuttals, I agree with the reviewer "RLfS" and the authors present a good rebuttal. I will raise my score to 6 in the final recommendation.

---

### Author Response · Authors · 2021-08-10
**General Response**

We thank the reviewers for their time and very helpful feedback. We are very happy to see that they generally found our work to be novel and a clear improvement over the state-of-the-art.

We note that the reviewers’ suggestions generally fall into two categories: comments regarding the clarity of the paper, and comments about our experimental analysis. We address both below.

**Clarity**

The reviewers identified a few typos and provided detailed suggestions on how to improve the presentation of our mathematical notation. We are very grateful for their help and have integrated their suggestions into the paper. Specifically, all identified typos have been corrected, the undefined quantities in Lemma 1 are now given a separate definition, and Equation 1 is now correctly expressed as an argmax of the parameters $\theta$.

Reviewer Bp9h suggests a flow chart to illustrate our proposed method. We had such a figure created already at time of submission, and only left it out due to space constraints. We will include it in the camera-ready version, making use of the additional page allowance.

We once again thank the reviewers for helping to improve the clarity of our paper. It is now much improved. Unfortunately, we can not upload a revised manuscript during the author response period but the changes were straightforward.

**Evaluation**

Reviewer RLfS asks why we did not use the 3 datasets utilized by the original Recurrent Classifier Chain paper [15]. These datasets are designed for a setting in which subset accuracy is inappropriate; namely,   Extreme Multi-Label Classification (EMLC) which feature very large label sets in hundreds. One reason is that EMLC requires learning many infrequent tail labels, which subset accuracy is unsuited for as it weighs all labels equally. Indeed, this is one of the few issues of [15] that their reviewers also had, based on their publicly available reviews [Paper 2802 Reviews, 2017]. To quote one of those reviewers for the original use of these datasets [1]: **“It is also a bit strange to optimize subset zero-one loss in MLC applications with hundreds of labels. Subset zero-one loss is for sure not the measure of interest in extreme MLC applications”**. For this reason, we choose multi-label datasets for which optimizing subset accuracy is appropriate.

Reviewer DcQS and JRCf suggest that we extend our experimental analysis on more datasets with varying numbers of labels. We have taken their advice and added additional 3 datasets to our analysis: Enron (53 classes) [2], EukaryoteGO (22 classes) [3], and Yeast (14 classes) [4].  Results (shown in Tables 1-3 below) indicate that our method continues to consistently outperform the state-of-the-art methods.


**<<Table 1: Performance of each method on the Enron dataset.>>**

| Metric | RBCC (Ours) | RCC | TS-RCC | OF-RCC | BCC | BD |
|:---:|:---:|:---:|:---:|:---:|:---:|:---:|
| Subset Acc. | **0.305**+/-0.016 | 0.100+/-0.001 | 0.090+/-0.005 | 0.1470.019 | 0.090+/-0.007 | 0.2570.020 |
| Hamming Loss | **0.027**+/-0.004 | 0.043+/-0.003 | 0.041+/-0.001 | 0.040+/-0.003 | 0.041+/-0.002 | 0.030+/-0.001 |
| Macro-F1 | **0.233**+/-0.013 | 0.0051 +/- 0.051 | 0.054+/-0.001 | 0.079+/-0.006 | 0.052+/-0.003 | 0.199+/-0.018 |
| Micro-F1 | **0.759**+/-0.075 | 0.607+/-0.020 | 0.614+/-0.023 | 0.617+/-0.022 | 0.611+/-0.022 | 0.716+/-0.012 |

**<<Table 2: Performance of each method on the EukaryoteGO dataset.>>**

| Metric | RBCC (Ours) | RCC | TS-RCC | OF-RCC | BCC | BD |
|:---:|:---:|:---:|:---:|:---:|:---:|:---:|
| Subset Acc. | **0.906+/-0.041** | 0.813+/-0.018 | 0.792+/-0.020 | 0.863+/-0.008 | 0.771+/-0.029 | 0.766+/-0.007 |
| Hamming Loss | **0.006+/-0.003** | 0.011+/-0.007 | 0.010+/-0.005 | 0.008+/0.001 | 0.008+/-0.002 | 0.013+/-0.000 |
| Macro-F1 | **0.907+/-0.053** | 0.600+/-0.009 | 0.590+/-0.023 | 0.629+/-0.014 | 0.537+/-0.020 | 0.466+/-0.028 |
| Micro-F1 | **0.937+/-0.028** | 0.892+/-0.024 | 0.863+/-0.030 | 0.921+/-0.005 | 0.843+/-0.025 | 0.872+/-0.006 |


**<<Table 3: Performance of each method on the Yeast dataset.>>**

| Metric | RBCC (Ours) | RCC | TS-RCC | OF-RCC | BCC | BD |
|:---:|:---:|:---:|:---:|:---:|:---:|:---:|
| Subset Acc. | **0.243**+/-0.016 | 0.150+/-0.008 | 0.148+/-0.011 | 0.219+/-0.022 | 0.153+/-0.013 | 0.121+/-0.014 |
| Hamming Loss | 0.195+/-0.008 | 0.254+/-0.006 | 0.257+/-0.005 | **0.188**+/-0.005 | 0.248+/-0.010 | 0.203+/-0.008 |
| Macro-F1 | **0.420**+/-0.015 | 0.282+/-0.053 | 0.279+/-0.050 | 0.394+/-0.010 | 0.291+/-0.055 | 0.293+/-0.012 |
| Micro-F1 | 0.658+/-0.025 | 0.528+/-0.021 | 0.530+/-0.019 | **0.682**+/-0.012 | 0.526+/-0.024 | 0.602+/-0.022 |

-----

References

Paper 2802 Reviews (Maximizing Subset Accuracy with Recurrent Neural Networks): https://proceedings.neurips.cc/paper/2017/file/2eb5657d37f474e4c4cf01e4882b8962-Reviews.html

[2] Klimt, B.; Yang, Y. (2004). The Enron Corpus: A New Dataset for Email Classification Research. In Proc. ECML04, Pisa, Italy, 217--226.

[3] Xu, Jianhua; Liu, Jiali; Yin, Jing; Sun, Chengyu (2016). A multi-label feature extraction algorithm via maximizing feature variance and feature-label dependence simultaneously. In Knowledge-Based Systems, 98(), 172--184.

[4] Elisseeff, A.; Weston, J. (2001). A Kernel Method for Multi-Labelled Classification. In Advances in Neural Information Processing Systems, 681--687.

---

### Decision · Program_Chairs · 2021-09-27

**Decision:**

Accept (Poster)

**Comment:**

The algorithm introduced by the authors is sound and obtains promising results as given in the experimental results. The main problems pointed out by reviewers are its clarity and readability. All those problems are mainly caused by the used notation and typos in equations. Nevertheless, the authors were very responsive, clarified almost all issues, and shared with reviewers intended changes to be incorporated into the final version of the paper. Taking this into account we lean toward acceptance of the paper.

Additional comments:
- Please also correct text and equations in lines 217-218: I suppose it should be $[0, 1]^L$ instead of $\\{0,1\\}^L$; it is better to define a loss function using all its parameters (true labels and predicted probabilities).